# Astemizole Sensitizes Adrenocortical Carcinoma Cells to Doxorubicin by Inhibiting Patched Drug Efflux Activity

**DOI:** 10.3390/biomedicines8080251

**Published:** 2020-07-29

**Authors:** Anida Hasanovic, Méliné Simsir, Frank S. Choveau, Enzo Lalli, Isabelle Mus-Veteau

**Affiliations:** Université Côte d’Azur, CNRS, Institut de Pharmacologie Moléculaire et Cellulaire, 06560 Valbonne, France; anida160@hotmail.com (A.H.); simsir@ipmc.cnrs.fr (M.S.); choveaufrank@gmail.com (F.S.C.); lalli@ipmc.cnrs.fr (E.L.)

**Keywords:** adrenocortical carcinoma, Patched, drug efflux pump, chemotherapy resistance, small lead molecule, repositioning, cancer therapy

## Abstract

Adrenocortical carcinoma (ACC) presents a high risk of relapse and metastases with outcomes not improving despite extensive research and new targeted therapies. We recently showed that the Hedgehog receptor Patched is expressed in ACC, where it strongly contributes to doxorubicin efflux and treatment resistance. Here, we report the identification of a new inhibitor of Patched drug efflux, the anti-histaminergic drug astemizole. We show that astemizole enhances the cytotoxic, proapoptotic, antiproliferative and anticlonogenic effects of doxorubicin on ACC cells at concentrations of astemizole or doxorubicin that are not effective by themselves. Our results suggest that a low concentration of astemizole sensitizes ACC cells to doxorubicin, which is a component of the standard treatment for ACC composed of etoposide, doxorubicin, cisplatin and mitotane (EDPM). Patched uses the proton motive force to efflux drugs. This makes its function specific to cancer cells, thereby avoiding toxicity issues that are commonly observed with inhibitors of ABC multidrug transporters. Our data provide strong evidence that the use of astemizole or a derivative in combination with EDPM could be a promising therapeutic option for ACC by increasing the treatment effectiveness at lower doses of EDPM, which would reduce the severe side effects of this regimen.

## 1. Introduction

Adrenocortical carcinoma (ACC) is a rare endocrine malignancy with an annual incidence of 0.5–2 cases per million people per year [1,2]. The prognosis of ACC is overall poor, particularly with cortisol-producing tumors [3]. Complete tumor removal by surgery is the most important step in the management of patients with primary, recurrent or metastatic ACC. However, 40–50% of patients who underwent surgery have a survival rate of about five years [4]. The main reason for that is the advanced stage of the disease at the moment of diagnosis, which makes impossible the complete removal of the tumor with surgery, and relapse appears in approximately 70–80% of patients after resection. ACC patients present a high risk of relapse and metastases even when the primary tumor is diagnosed and surgically excised at an early stage. Outcomes did not improve despite extensive research [5]. Mitotane and etoposide/doxorubicin/cisplatin chemotherapy have been validated for improved recurrence-free survival of patients with ACC [6]. However, a minority of patients experience a sustained benefit of this regimen which presents side effects so severe that often makes the patients unable to attain target drug doses shown to give a survival benefit [7,8]. Five-year survival rates of 35% in advanced ACC suggest that resistance to adjuvant therapy is the main factor behind treatment failure [9]. Therefore, it is necessary to find new therapeutic options to improve ACC treatment outcomes and increase the overall survival of patients.

It has long been postulated that the multidrug ABC transporter P-glycoprotein (P-gp/ABCB1/MDR1) mediates the main mechanism of resistance within cancer cells [10]. However, we recently showed that the Hedgehog receptor Patched (Ptch1), which is overexpressed in many recurrent and metastatic cancers ([11,12], and Human Protein Atlas website http://www.proteinatlas.org/ENSG00000185920-PTCH1/cancer), is also expressed in ACC and contributes to resistance of ACC to treatment [13]. Indeed, Ptch1 pumps chemotherapeutic agents such as doxorubicin (dxr) out of cancer cells, leading to chemotherapy resistance [13,14]. Ptch1 is not part of the ABC transporters family but uses the proton motive force to efflux drugs similar to the bacterial efflux pumps from the RND family [15]. This may seem surprising; however, the high glucose utilization and the alteration of energy metabolism that occur in cancer cells have been shown to lead to lactate production which is transported out of the cells and acidifies the extracellular medium [16]. This pattern of the acidic extracellular environment and alkaline cytosol is considered as a hallmark of malignant cancers and is referred to as a “reversed pH gradient” [17]. Accordingly, Ptch1 functions as an efflux pump only in cancer cells. This makes Ptch1 a particularly relevant therapeutic target, and Ptch1 drug efflux inhibitors particularly interesting due to their specificity for cancer cells.

We developed screening tests to identify molecules that inhibit the resistance to doxorubicin (dxr), a chemotherapeutic agent used to treat many cancers, conferred by human Ptch1 to yeast, and the efflux of dxr by Ptch1 [18]. This led to the discovery of two Ptch1 inhibitors. The first, panicein A hydroquinone (PAH), a compound purified from a marine sponge, increased the cytotoxicity of dxr against melanoma cells in vitro and in vivo [19,20]. The second inhibitor, methiothepin, a drug-like compound from the Prestwick Chemical library, increased the efficacy of dxr against adrenocortical carcinoma cells in vitro and in vivo [13]. When screening the Prestwick Chemical library, we identified a second compound able to inhibit the resistance to dxr conferred by Ptch1 to yeast and the efflux of dxr mediated by Ptch1. This compound is astemizole, a non-sedating anti-histaminergic drug. In the present study, we report results showing that astemizole is a Ptch1 drug efflux inhibitor and the effects of this compound on the response of ACC cells to doxorubicin.

## 2. Materials and Methods

### 2.1. Chemical and Biological Material

Astemizole was purchased from Santa Cruz: CAS number: 68844-77-9; MW: 458.57 g/mol; molecular formula: C_28_H_31_FN_4_O. Doxorubicin hydrochloride was purchased from Sigma-Aldrich. BODIPY-cholesterol was purchased from Avanti (Topfluor, Avanti Polar Lipids Inc, Alabaster, AL, USA).

*Saccharomyces cerevisiae* strain K699 (Mata, ura3 and leu 2–3, kindly donated by R. Arkowitz) was transformed with the expression vectors pYEP-hPtc-MAP (human Ptch1) or pYEP-hSmo-MAP (control), and grown at 18 °C until OD_600_ 5–7 as previously described [20].

The human adrenocortical carcinoma cell line H295R was grown in DMEM/F12 medium containing 2% NuSerum (BD), 1% ITS+ (BD) and penicillin/streptomycin (Invitrogen, Thermo Fisher Scientific, Illkirch, France) at 37 °C and 5% CO_2_ as previously described [13]. This medium was supplemented with 20 µg/µL EGF and 20 µg/µL FGF for spheroid 3D culture.

Human embryonic kidney cells HEK293 were grown in DMEM medium (Gibco, Thermo Fisher Scientific, Illkirch, France) supplemented with 10% fetal calf serum (Thermo Fisher Scientific, Illkirch, France) and 1% penicillin-streptomycin (Gibco, Thermo Fisher Scientific, Illkirch, France) at 37 °C in 5% CO_2_ water-saturated atmosphere.

### 2.2. Screening on the Resistance to Doxorubicin of Yeast Expressing Human Ptch1

The screening of molecules on yeast was carried out as described in [18]. Briefly, *S. cerevisiae* expressing human Ptch1 were grown at 30 °C up to an OD_600_ between 1 and 2in minimal medium containing 2% of glucose and amino acid cocktail without leucine, and then diluted in rich medium containing 2% glucose in 96-well plates. An amount of 10 µM of molecules to be tested was added in 8 wells (10 molecules can be tested on one plate) and 10 µM of dxr was added to half of the wells. Plates were incubated at 18 °C on a shaker at 1250 rpm (microtiter plate shaker SSL5 Stuart) and absorbance at 600 nm was recorded for about 72 h.

### 2.3. Cytotoxicity Assays

Cytotoxicity assays on H295R cells were carried out as previously described [13]. Briefly, H295R cells were grown in 96-well plates to achieve 70% to 80% confluence. The medium was then replaced with 100 µL/well of medium containing astemizole or DMSO as control for 2 h. An amount of 100 µL of medium containing increasing concentrations of dxr was then added. For astemizole IC_50_ calculation, cells were incubated with increasing concentrations of astemizole. Plates were incubated at 37 °C and 5% CO_2_. After 48 h, cells were incubated 3 h at 37 °C with neutral red (NR) solution following the manufacturer’s protocol. Absorbance was measured in a microplate reader (Multiskan Go Microplate Spectrophotometer from Thermo Fisher Scientific, Illkirch, France). IC_50_ corresponding to the concentration of dxr or astemizole that induced a 50% decrease in the number of live cells was calculated using the GraphPad Prism 6 software.

### 2.4. Synergy Analysis of Drug Combination

Synergy analysis was performed using the Chou-Talalay method for drug combination as described in [21,22]. Analysis was done on the basis of dose–response curves of cells treated for 48 h with dxr alone, astemizole alone and dxr in the presence of 1 µM astemizole. Normalized isobolograms and Chou-Talalay’s plots (Fa-CI plot were Fa is the fraction affected and CI is the combination index) were created using the Compusyn Software (Version 1.0 downloaded from www.combosyn.com [22]). These allow quantitative determination of drug interactions, where CI < 1 (below the diagonal), =1 (diagonal) and >1 (above the diagonal) indicate synergism, additive effect and antagonism, respectively. Dose-reduction index (DRI) is reported.

### 2.5. Apoptosis Measurements

Cells were seeded at a density of 7000 cells per well in a 96-well white polystyrene plate (Falcon Corning 96 Well Plate) in triplicate and cultured overnight at 37 °C and 5% CO_2_. After removal of the medium, cells were treated 48 h with medium containing DMSO, dxr alone, astemizole alone or dxr and astemizole together. Quantification of caspase 3/7 activity was performed using the luminescent assay Caspase-Glo 3/7 (Promega, Charbonnières-les-Bains, France) and a luminometer (Glomax 96 Microplate Luminometer from Promega, Charbonnières-les-Bains, France) following the manufacturer’s protocol.

### 2.6. Proliferation

Cells were seeded at a density of 5000 cells per well in 96-well plates (Falcon 96 Well Clear Microplate, Corning Inc, Corning, NY, USA) in triplicate and grown for 24 h at 37 °C and 5% CO_2_. After removal of the medium, cells were treated with medium containing serial dilutions of dxr in the presence or the absence of astemizole. DMSO was added to the control wells. After 7 days at 37 °C and 5% CO_2_, the NR test was used for the quantification of living cells. IC_50_ values were calculated using the GraphPad Prism 6 software.

### 2.7. Clone Formation

Effect of astemizole on H295R clone formation was carried out as previously described [13]. A total of 5000 H295R cells were seeded per well in 24-well plates (Falcon from Corning Inc, Corning, NY, USA), treated in triplicate with DMSO as control, astemizole alone, dxr alone or a combination of astemizole and dxr, and incubated at 37 °C and 5% CO_2_. After 14 days, clones were stained with crystal violet and pictures were taken. Cells were then solubilized and absorbance was measured in a microplate reader at 550 nm (Multiskan Go Microplate Spectrophotometer, Thermo Scientific, Illkirch, France).

### 2.8. Spheroid Formation in 3D Culture

H295R cells were cultured in DMEM/F-12, 2% NuSerum, 1% ITS Plus and penicillin/streptomycin supplemented with 20 µg/µL EGF and 20 µg/µL FGF. Cells were seeded in 96-well plates (Falcon from Corning Inc, Corning, NY, USA) at a density of 5000 cells per well in triplicate, treated with DMSO as control, astemizole alone, dxr alone or a combination of astemizole and dxr, and incubated at 37 °C and 5% CO_2_. After 14 days, images of each well were taken using a cell imaging multi-mode reader (Cytation 5 from Bio Tek, Colmar, France). Then, 100 µL of the medium was removed from each well and 100 µL of Cell Titer Glow 3D reagent (Promega) was added. Spheroids were disrupted according to the protocol from the Cell Titer Glow 3D Viability Assay (Promega) and viability was measured in a microplate reader at 550 nm (Multiskan Go Microplate Spectrophotometer, Thermo Scientific).

### 2.9. SDS-PAGE and Western Blotting

Western blots were carried out as previously described [13]. Total RIPA extracts from cells were prepared and protein concentration was estimated using the DC Protein Assay (Bio-Rad, Feldkirchen, Germany). An amount of 50 to 80 µg of proteins was separated on SDS-PAGE and transferred to membranes of nitrocellulose (Amersham, Bath, UK). After 1 h blocking in 5% non-fat milk, membranes were incubated overnight at 4 °C with rabbit anti-hPtch1 antibody (Abcam ab53715; 1/1000) or mouse anti-β-tubulin antibody (Sigma Aldrich, Merck, Darmstadt, Germany; 1/1000). After washing, membranes were incubated for 45 min with anti-rabbit (1:2000) or anti-mouse (1:5000) immunoglobulin coupled to horseradish peroxidase (Dako-Agilent, Santa Clara, CA, USA). ECL Prime Western Blotting detection reagent (Amersham) on a Fusion FX imager (Vilber Lourmat, Collegien, France) was used for detection.

### 2.10. Efflux Measurements

Efflux measurements were carried out as described in [19]. On yeast: hPtch1-expressing yeasts or control yeasts were grown to an OD_600_ of 5, centrifuged, washed with cold water and resuspended in HEPES–NaOH buffer (pH 7.0) containing 5 mM 2-deoxy-D-glucose and 10 µM dxr for 2 h at 4 °C. After centrifugation, one sample was immediately fixed with 4% PFA for dxr loading control, while the other samples were resuspended in HEPES–NaOH buffer (pH 7.0) containing 5 mM 2-deoxy-D-glucose and DMSO or 10 µM astemizole, and incubated 10 min at 20 °C with gentle shaking in a Benchmark Multi-therm shaker (Innova 2000, New Brunswick Scientific, Enfield, CT, US) protected from light. Yeasts were centrifuged, resuspended in 4% PFA and deposited on a coverslip with SlowFade Gold antifade reagent containing DAPI (Invitrogen).

On cells: After seeding on coverslips in 24-well plates, H295R cells were incubated at 37 °C and 5% CO2 with 10 μM doxorubicin (dxr) or BODIPY-cholesterol (Bo-chol) in physiological buffer (140 mM NaCl, 5 mM KCl, 1 mM CaCl_2_, 1 mM MgSO_4_, 5 mM glucose, 20 mM HEPES, pH 7.4). After 2 h, some coverslips were immediately fixed with PFA and mounted in SlowFade Gold antifade reagent containing DAPI (Invitrogen) for loading control. The other coverslips (a triplicate per condition) were incubated with physiological buffer containing DMSO or 10 µM astemizole for 30 min under gentle shaking at room temperature and protected from light, fixed with PFA and mounted as described above.

Images were acquired with a Zeiss Axioplan 2 fluorescence microscope coupled to a digital charge-coupled device camera using a 63X objective for yeast or a 40X /1.3 Plan NeoFluar objective for cells, and filters for Alexa 594 or FITC for dxr or Bo-chol analysis, respectively. Dxr or Bo-chol fluorescence was quantified using the ImageJ software. About 100 yeast/cells (from three wells) were scored per condition.

### 2.11. Electrophysiology

Transfection and experiments were carried out as described in [23]. HEK293 cells were plated onto 35 mm dishes for transfection with pSI-hERG, and experiments were performed over the following 1–2 days. In the standard bath solution, pulled pipettes had resistances of 2–4 MΩ when filled with intracellular solution containing 74.5 mM KCl, 70.5 mM K-aspartate, 5 mM HEPES, 2 mM EGTA, 2 mM K_2_ATP and 0.3 mM MgCl_2_ pH 7.2 with KOH. Whole-cell membrane currents were measured and filtered at 3 kHz using a patch-clamp amplifier RK 400 (Bio-Logic Science Instruments, Seyssinet-Pariset, France), digitized at 10 kHz with the 12-bit analog-to-digital converter Digidata-1322 (Axon Instrument, Sunnyvale, CA, United States) and recorded with the software Clampex 8.2 (Axon Instrument). hERG currents in HEK cells were recorded using an external solution composed of: 145 mM NaCl, 4 mM KCl, 1 mM MgCl_2_, 1 mM CaCl_2_, 5 mM HEPES and 5 mM glucose, pH adjusted to 7.4 with NaOH and flowed at 1–2 mL/min through the dish. Astemizole was dissolved in DMSO to produce a 50-mM stock solution stored at −20 °C. The final experimental concentration was reached by dilution in the locally perfused external medium described above. Experiments were performed at room temperature.

### 2.12. In Silico Docking

Docking of astemizole on the Ptch1 structure was performed using the Vina toolkit [24] in USCF Chimera [25] as previously described [20]. The structure of Ptch1 (PDB ID: 6N7H, chain A) [26] was prepared using USCF Chimera Predock Toolkit, the Dunbrack rotamer 2010 library [27] for the missing side chains and ANTECHAMBER Amber ff14SB force field [28] for charges assignment. The docking was done by targeting the central cholesterol cavity. The 10 best poses of the docking (ranked by score) were used for the analysis. Interactions were analyzed using PoseView [29], a function of the Proteins*Plus* web server [30]. Amino acids within a radius of 6 Å from astemizole were selected and compared to amino acids within a radius of 6 Å from cholesterol.

Sequence alignment was performed on 30 sequences from Patched family members using the T-COFFEE server for transmembrane proteins [31] to identify conserved amino acids.

### 2.13. Profiling and Pharmacokinetic Studies

Profiling and PK studies were performed by GalenAuxi Co (Oxford, UK) as previously described [13]. Astemizole was formulated in DMSO/Solutol^®^ HS15/ PBS and administered to non-fasted male ICR mice (weighing 20–30 g) intravenously (IV) at the dosing volume of 5 mL/kg and orally (PO) at 10 mL/kg, 3 mice per time point. Plasma samples (50 μL) were collected via the facial vein at 3, 10, 30, 60, 120, 240, 360 and 1440 min post-dose for the IV group, and at 10, 30, 60, 120, 240, 360, 480 and 1440 min post-dose for the PO group in K_2_EDTA-coated tubes, mixed gently and centrifuged at 2500 g for 15 min at 4 °C within 1 h after collection. For control animals, blood was collected by cardiac puncture. Plasma samples were prepared in acetonitrile for LC-MS/MS analysis. Peak areas were recorded and the concentrations of the test compound in the unknown plasma samples were determined using the respective calibration curve. Plasma concentration–time curves were constructed and pharmacokinetic parameters of astemizole were obtained from the analysis of the plasma data.

### 2.14. Statistical Analysis

All results presented were obtained from at least three independent experiments, and data are given as the mean value ± SEM. The GraphPad Prism 6 software (San Diego, CA, US) was used for statistical analyses with one-way analysis of variance (ANOVA) followed by Bonferroni’s multiple comparison tests.

## 3. Results

### 3.1. Identification of a New Inhibitor of the Resistance to Doxorubicin and the Doxorubicin Efflux Conferred by Ptch1 to Yeast

We previously reported that yeast expressing human Patched (hPtch1) were able to grow in the presence of chemotherapeutic agents such as doxorubicin (dxr) and to efflux dxr out of cells using the proton motive force [14]. To find inhibitors of Patched drug efflux activity, we developed a screening test based on the growth of hPtch1-expressing yeast in the presence of dxr [18]. We measured the effect of 1200 compounds from the Prestwick Chemical drug library on the growth of hPtch1-expressing yeast in the presence of dxr, and we discovered that the methiothepin maleate (P375) significantly inhibited the resistance of hPtch1-expressing yeast to dxr in contrast to the other compounds such as P298 that did not affect it (Figure 1A). During this screening, we observed that another drug from the same library, P136, was also able to inhibit the growth of hPtch1-expressing yeast in the presence of dxr without effects on the growth of hPtch1-expressing yeast in the absence of dxr (Figure 1A), suggesting that this molecule inhibited specifically dxr resistance conferred by hPtch1 to yeast.

Then, we measured the effect of P136 on the amount of dxr in hPtch1-expressing yeast by fluorescence microscopy using the natural fluorescence of dxr as described before [19]. Experiments were performed in the presence of 2-deoxy-D-glucose to inhibit ATP production and thus the contribution of yeast MDR transporters from the ATP-binding cassette (ABC) to the drx efflux. This allowed us to better visualize Ptch1’s contribution to the dxr efflux because Patched does not use ATP hydrolysis but rather the proton motive force to get the drugs out of the cells. After incubation with dxr, we observed that dxr accumulated in yeast. After removing the medium containing dxr and 10 min in buffer containing DMSO (P136 solvent), hPtch1-expressing yeasts contain significantly less dxr fluorescence than control yeasts, consistently with the dxr efflux activity of hPtch1 previously reported [14] (Figure 1B). Remarkably, the presence of P136 in the efflux buffer increases the amount of dxr in hPtch1-expressing yeast significantly more than in control yeast (Figure 1B), indicating that P136 specifically inhibited dxr efflux activity mediated by hPtch1.

Compound P136 corresponds to astemizole, a well-known non-sedating second-generation H_1_-histamine receptor antagonist.

### 3.2. Astemizole Increases the Cytotoxicity of ACC Standard Treatment

We recently showed that ACC expresses Ptch1 and that the first Ptch1 drug efflux inhibitor identified from our screening of the Prestwick Chemical library, methiothepin, was able to increase the efficacy of doxorubicin against the human ACC cell line H295R [13]. Therefore, we wanted to know if astemizole was also able to increase dxr efficacy against ACC cells. H295R cells were grown to 80% confluence and incubated with increasing concentrations of dxr in the absence or the presence of 1 µM astemizole for 48 h. The cell viability assay using neutral red revealed that a dose of 1 µM astemizole, that was not toxic to ACC cells by itself, increased the cytotoxicity of dxr by about 10 times (Figure 2A, Table 1). The gold standard treatment given to ACC patients is composed of a mixture of doxorubicin, etoposide, cisplatin and mitotane (EDPM). Interestingly, we observed that astemizole increased about eight times the efficacy of dxr when combined with 10 µM etoposide and 10 µM mitotane (EDM) (Figure 2B, Table 1).

The Chou-Talalay’s plot (Fa-CI plot) and normalized isobologram analysis clearly showed that dxr and astemizole exhibit a significant synergism as demonstrated by the cooperativity indices (CI) far below 1 (Appendix A). Similar results were obtained for the combination of astemizole with EDM treatment (Appendix A).

Western blot analysis and quantification of the Ptch1 protein amount relative to β-tubulin revealed that astemizole treatment did not affect Ptch1 protein expression (Figure 2C).

### 3.3. Astemizole Increases the Pro-Apoptotic, Anti-Proliferative and Anti-Clonogenic Effects of Doxorubicin

We observed that the presence of astemizole in the growth medium increased the anti-proliferative effect of doxorubicin on H295R cells. In the presence of 0.5 μM astemizole, which does not affect cell viability by itself, the dxr-IC_50_ decrease was about three times (Figure 3A).

Apoptosis experiments based on caspase 3/7 activation measurements indicated that the addition of astemizole to dxr treatment increased the number of apoptotic cells in a dose-dependent manner. Indeed, the use of 0.5 µM of astemizole, which does not affect caspase 3/7 activity by itself, multiplies it by five the number of apoptotic cells relative to cells treated with 2 µM dxr without astemizole (Figure 3B).

We also observed that the combination of dxr and astemizole significantly inhibited the ability of H295R cells to form clones at a concentration of dxr that did not affect it by itself, and increased four times the anti-clonogenic activity of dxr (Figure 3C).

H295R cells form spheres when grown in 3D culture medium. ACC cells were grown in these conditions in the presence of increasing concentrations of dxr, astemizole or a combination of dxr + astemizole. Figure 3D shows that the spheroids formed after 14 days were significantly smaller when treated with the combination of dxr + astemizole, while the concentration of dxr or astemizole used did not affect spheroid growth by itself.

Our results clearly show that astemizole significantly increased the anti-proliferative, pro-apoptotic and anti-clonogenic effects of dxr against ACC cells.

### 3.4. Astemizole Inhibits Doxorubicin Efflux from ACC Cells

We recently showed that the inhibition of Ptch1 expression using siRNA strongly reduced the efflux of dxr from H295R cells revealing that Ptch1 participated in dxr efflux and drug resistance in these cells [13]. We observed that treatment of H295R cells with astemizole for 48 h had no effects on Ptch1 protein expression (Figure 2C), indicating that the increase in dxr cytotoxicity observed in the presence of astemizole is not due to the inhibition of Ptch1 expression but rather to the inhibition of Ptch1 drug efflux activity.

As exemplified in Figure 4A, incubation of H295R cells with dxr for 2 h induced a strong accumulation of dxr in the cells. The intracellular dxr amount was drastically reduced after 30 min in the efflux buffer. Quantification of dxr fluorescence in cells indicated that 60 to 70% of dxr was transported out of the cells over this period time. Interestingly, the presence of astemizole in the efflux buffer allowed keeping about 73% of dxr inside the cells. This result demonstrates that astemizole strongly inhibited the efflux of dxr from H295R cells.

Remarkably, we observed that astemizole also inhibits dxr efflux from melanoma cells such as MeWo which endogenously express Ptch1 [20], and increases dxr cytotoxicity in these cells (Sup. Figure 2A,B).

### 3.5. Astemizole Inhibits Cholesterol Efflux from ACC Cells

We have previously shown that Ptch1 regulates Hedgehog signaling by transporting cholesterol out of cells [32]. Thus, we measured the effect of astemizole on the cholesterol efflux in ACC cells. To do so, we used a fluorescent derivative of cholesterol: BODIPY-cholesterol (Bo-chol). As shown in Figure 4B, incubation of cells with Bo-chol induced a strong accumulation of Bo-chol inside the cells. After 30 min in the efflux buffer, we observed a reduction of 29% in Bo-chol fluorescence inside the cells, indicating an efflux of Bo-chol. The presence of astemizole in the efflux buffer allowed keeping about 95% of Bo-chol inside the cells. This result strongly suggests that astemizole also inhibited cholesterol efflux in ACC cells.

### 3.6. Astemizole Binds to Ptch1 in the Same Binding Pocket as Cholesterol and Doxorubicin

We performed in silico docking on the cryo-EM structure of Ptch1 PDB ID 6N7H [26]. The available structures to date have up to three cavities containing cholesterol. The one we are particularly interested in is the central cavity. This central cavity is not found in all structures, suggesting that it is a dynamic cavity within the transport mechanism of cholesterol. This is supported by the fact that it is constructed between loops, which therefore demonstrates flexibility and ability to accommodate different sized ligands. The presence of numerous aromatic amino acids with polar groups such as tyrosine and tryptophan, as well as polar residues among the hydrophobic ones, also gives the ability to accommodate various ligands and provide a wide range of interactions.

In view of the potential major role of this cavity in Ptch1 efflux activity, we have looked at whether some of the amino acids that compose it can be the cause of pathology in the case of mutation, which would underline their role in the transport of cholesterol. This is the case for eight of them according to the BioMuta database [33] (Table 2, amino acids in bold). If an amino acid is vital to the proper functioning of a protein, one would expect it to be conserved within the family protein. Among those surrounding the cholesterol, five are conserved in the Patched family (between * in Table 2), two of which have side chains directed to the cholesterol (Leu427 and Ala497).

All these elements comfort us in carrying out our docking on this cavity specifically. The analysis was performed on the 10 best poses of the docking (ranked by score). Since both the amino acids involved in the interaction with astemizole and the nature of the interactions were similar among those poses, only one pose is presented (Figure 5).

We previously observed that the best docking poses for dxr superimposed on cholesterol, and, very interestingly, we show here that the best docking poses for astemizole superimposed with cholesterol and dxr. As presented in Table 2, 26 amino acids over the 33 surrounding cholesterols and 10 amino acids over the 16 surrounding dxrs in a radius of 6 Å are also in a radius of 6 Å of astemizole (Table 2). Interestingly, three amino acids for which mutations are responsible for diseases, suggesting that they are important for Ptch1 function, are in the surrounding of cholesterol, dxr and astemizole: L128, D776, W1018.

From the 2D representation of the interaction between astemizole and Ptch1, we notice that most of them are hydrophobic interactions (W129, L777, I780, W1018) and one is a pi-stacking type (F1017). With this docking being performed on a rigid structure, it does not unveil all interactions astemizole can have with Ptch1. It is highly possible that the polar atoms of astemizole are engaged in polar interactions such as H bonds but are not found or represented in this in silico docking.

### 3.7. Astemizole is Cytotoxic for ACC Cells

Astemizole has gained great interest as a potential anticancer drug because it targets several proteins involved in cancer including the Eag1 (*ether à-go-go-1*) potassium channel that is overexpressed in many cancers and is strongly related to carcinogenesis and tumor progression [34,35]. We, therefore, tested the effect of astemizole on H295R cells, and we observed that astemizole was cytotoxic for ACC cells with an IC_50_ of about 7 µM (Figure 6A) and inhibited their proliferation (Figure 6B). A similar effect was observed on melanoma cells, while astemizole only has a very slight effect on keratinocyte viability (Appendix A).

### 3.8. Astemizole Properties

Numerous structurally and functionally unrelated drugs block the hERG potassium channel, well known for its role in repolarizing the cardiac action potential. The alteration of hERG by pharmacological inhibition produces long QT syndrome and the lethal cardiac arrhythmia torsade de pointes. Astemizole has been withdrawn from the market because of QT prolongation [35,36,37]. Therefore, we measured the effect of astemizole on the hERG channel current by electrophysiology. I_hERG_ “pulse” amplitude was monitored during repetitive application of the protocol presented in Figure 7, and we observed that 1µM astemizole induced about a 70% reduction in I_hERG_ pulse amplitude, confirming the ability of astemizole to inhibit the hERG channel.

We also performed an ADME analysis on astemizole (Table 3). We determined its lipophilicity described by the distribution constant LogD, its solubility, which is an important parameter since water is the solvent of choice for the liquid pharmaceutical formulation, its metabolic stability measured in the presence of rat and human microsomes and NADPH, and its ability to bind to proteins. Results showed that astemizole is metabolically stable and possesses lipophilicity and solubility properties compatible with a drug candidate.

## 4. Discussion

In this study, we report that astemizole inhibited doxorubicin efflux in hPtch1-expressing yeast and alleviated resistance to dxr conferred by hPtch1 to yeast, while it had no effects on control yeast (Figure 1). In silico docking showed that the best poses for astemizole were in the central cholesterol-binding pocket of Ptch1, where we also found the best poses for doxorubicin (Figure 5, Table 2). These results suggest that astemizole could inhibit the efflux activity of the Hh receptor Ptch1 by direct binding to the Ptch1 protein.

We previously reported that Ptch1 is expressed in ACC and contributes to the resistance of these tumors to treatment [13]. We then tested astemizole on ACC cells and observed that this molecule, at a concentration that does not affect by itself, significantly increased the cytotoxic, pro-apoptotic, anti-proliferative and anti-clonogenic effects of doxorubicin, which is one of the chemotherapeutic agents of the standard treatment for ACC patients (Figure 2 and Figure 3). Our experiments revealed that astemizole was able to inhibit dxr efflux in ACC cells (Figure 5), in good agreement with results obtained in hPtch1-expressing yeast, explaining that astemizole increased dxr efficacy against ACC cells. Our results also showed that astemizole inhibited cholesterol efflux, which is the physiological activity of Ptch1. These results together with data obtained from hPtch1-expressing yeast and in silico docking strongly suggest that astemizole increased the sensitivity to doxorubicin of yeast and ACC cells by inhibiting the dxr efflux activity of Ptch1 and that Ptch1 is a new target of astemizole.

These data are in good agreement with previous studies reporting that when administrated in combination with a chemotherapeutic treatment, astemizole has shown a significant association with reduced mortality among cancer patients. One of the explanations given is that astemizole sensitizes cancer cells to chemotherapy and reverts multidrug resistance [38]. In vitro and preclinical studies suggest that astemizole may act synergistically in combination with chemotherapeutic agents. Indeed, astemizole was shown to synergistically potentiate the cytotoxicity of doxorubicin against doxorubicin-resistant human leukemia cells [39], and of gefitinib against human lung cancer [34]. This was attributed to an effect of astemizole on several proteins involved in cancer progression such as histamine receptors, ABC transporters and the potassium channels Eag1 and hERG [40]. However, our study suggests that these effects could also be due to the inhibition of the drug efflux activity of Ptch1 which is known to be expressed in many cancers (see [15] for review and the Human Protein Atlas website http://www.proteinatlas.org/ENSG00000185920-PTCH1/cancer). Astemizole is a second-generation H_1_ antihistamine drug approved in 1986 for the treatment of allergic rhinitis and conjunctivitis. It has a high affinity for the histamine H_1_ receptor, but also for the potassium channels Eag1 and hERG, as we have also reported in Figure 7, inducing serious adverse cardiac reactions. Accordingly, it was withdrawn from the market in 1999 for safety-related reasons [36]. Moreover, in vitro and nonclinical in vivo studies suggested that astemizole had anticancer properties by inhibiting the hERG channel and/or cytochrome CYP2J2, both of which being upregulated and overexpressed in various cancers [41]. We observed that astemizole also has a cytotoxic effect in ACC cells with an IC_50_ of about 7 µM (Figure 6), although overexpression of CYP2J2 or hERG in ACC has not been reported to date. We also observed that astemizole is cytotoxic on melanoma cells with an IC_50_ of about 7 µM, while its cytotoxicity is much lower with respect to non-tumorigenic keratinocytes (Sup. Figure 2). It is interesting to note that the concentration of astemizole shown to increase cytotoxicity and the anti-proliferative effects of doxorubicin (1 and 0.5 µM, respectively) have no effects in the absence of dxr (Figure 2, Figure 3 and Figure 6), indicating that astemizole acts synergistically in combination with dxr as already reported. This suggests that, at these low concentrations, astemizole acts only on Ptch1 drug efflux activity in these cells. We are currently working on the chemical modification of astemizole to make it lose its affinity for hERG. The optimized compound obtained will be tested on mice grafted with ACC cells to evaluate in vivo activity of this lead. The use of a combination of doxorubicin with a derivative of astemizole able to interact with Ptch1 but not with hERG would be a promising therapeutic option for ACC.

Our study provides strong evidence that astemizole is an inhibitor of the efflux activity of the Hedgehog receptor Ptch1, and that the synergistic effect of astemizole, when combined with chemotherapeutic agents such as doxorubicin, could also be related to the inhibition of Ptch1 drug efflux activity. Our results suggest that the use of a low dose of astemizole with no adverse effects by itself or of an astemizole derivative able to interact with Ptch1 but not with hERG could improve the effectiveness of the standard of care treatment for ACC patients, and could be a promising therapeutic option.

## Figures and Tables

**Figure 1 biomedicines-08-00251-f001:**
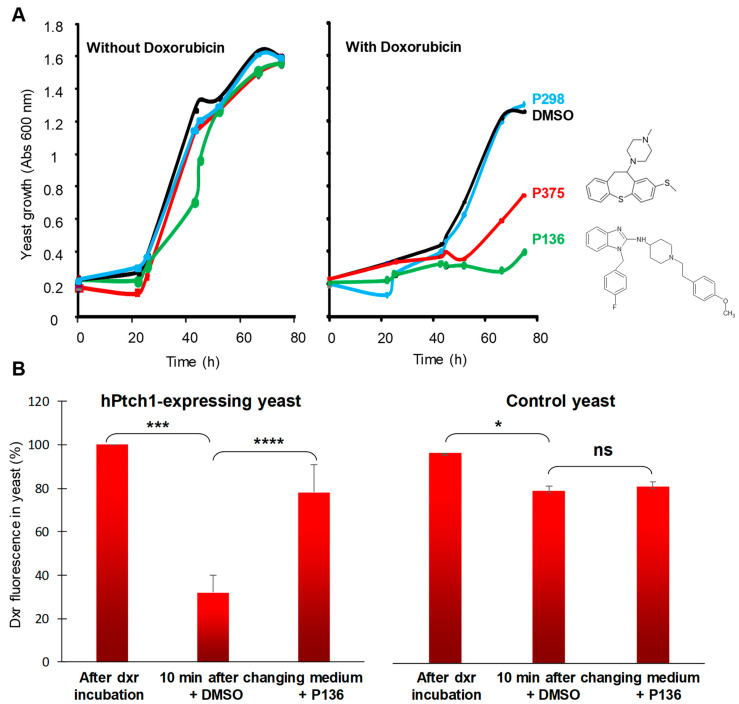
P136 inhibits doxorubicin efflux activity of hPtch1 when expressed in yeast. (**A**) P136 inhibits the resistance of hPtch1-expressing yeast to doxorubicin. hPtch1-expressing yeast were grown in the presence of 10 µM of P136, P298 or P375, and in the presence or absence of 10 µM doxorubicin (dxr). DMSO was used as control. The growth of yeast was measured by absorbance at 600 nm. (**B**) P136 inhibits the efflux of doxorubicin of hPtch1-expressing yeast. hPtch1-expressing yeast and control yeast were incubated with dxr for 2 h and fixed for dxr loading control or resuspended in buffer containing DMSO or 10 μM P136 for 10 min, fixed and deposited on a coverslip for analysis by fluorescence microscopy. The dxr fluorescence of 100 yeast for each condition from 3 independent experiments was quantified using the ImageJ software. Histograms represent the mean ± SEM and were analyzed using ANOVA multiple comparison test and Bonferroni correction. Significance is attained at *p* < 0.05 (*) (***: *p* < 0.0005, ****: *p* < 0.00005, ns: no significant difference).

**Figure 2 biomedicines-08-00251-f002:**
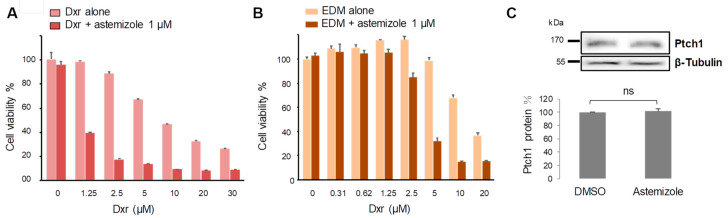
Astemizole increases the cytotoxic effect of ACC treatment. Cell viability was measured after 48 h treatment of H295R cells with serial dilutions of doxorubicin (dxr) (**A**) in the presence or the absence of 1 μM astemizole, (**B**) in the presence of 10 μM mitotane and 10 μM etoposide with or without 1 µM astemizole. The graphs reported are representative of more than 3 independent experiments. (**C**) Ptch1 protein expression after 48 h of treatment with 10 µM astemizole or DMSO as control. Quantification of Ptch1 expression level relative to β-tubulin was performed using the ImageJ software on 3 independent experiments. ns: no significant difference.

**Figure 3 biomedicines-08-00251-f003:**
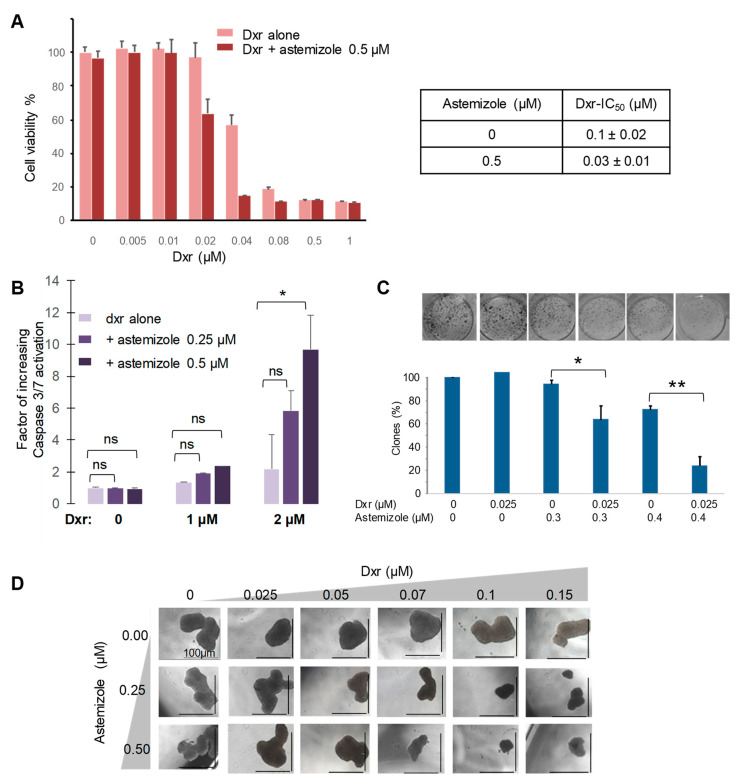
Astemizole increases the anti-proliferative, pro-apoptotic and anti-clonogenic effects of doxorubicin. (**A**) H295R cells were treated with serial dilutions of dxr with or without astemizole 0.5 µM for 7 days. The effect on cell proliferation was quantified using the neutral red assay. Data are represented as mean ± SEM. IC_50_ values were calculated from 3 different experiments using the Prism 6 software. (**B**) Apoptosis was evaluated using the luminescent assay Caspase-Glo 3/7 kit after 48 h incubation of H295R cells with medium alone or medium containing dxr alone, astemizole alone or a combination of dxr and astemizole. Caspase activation of cells was reported as a factor of the caspase activity of the control cells treated with DMSO. (**C**) For the clone formation test, H295R cells were treated with DMSO as a control, dxr or astemizole alone or a combination of dxr and astemizole. Clones were revealed with crystal violet solution after 14 days, pictures were taken and absorbance was read at 550 nm after solubilization. (**D**) H295R cells were plated in a 3D culture medium and treated with dxr or astemizole alone, or a combination of dxr and astemizole. After 14 days, pictures were taken using a cell imager. The experiment reported is representative of the 3 experiments performed. Histograms represent the mean ± SEM of 3 experiments. Data were analyzed using ANOVA multiple comparison test and Bonferroni correction. Significance is attained at *p* < 0.05 (*) (**: *p* < 0.005, ns: no significant difference).

**Figure 4 biomedicines-08-00251-f004:**
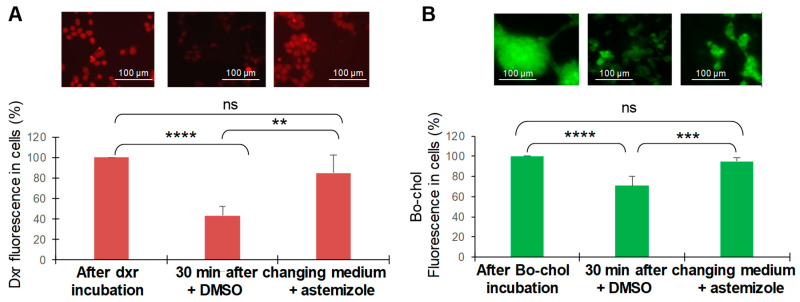
Astemizole inhibits the efflux activity of Ptch1. H295R cells were seeded on coverslips and incubated with dxr (**A**) or BODIPY-cholesterol (Bo-chol) (**B**). After 2 h, 3 coverslips were fixed for dxr or Bo-chol loading control. The other coverslips were incubated with DMSO or astemizole for 30 min and fixed. Images were acquired using fluorescence microscopy and a 40x objective. Fluorescence was quantified using the Image J software for 100 cells per condition. Histograms represent the mean ± SEM values of 3 independent experiments. Significance, calculated using ANOVA multiple comparison test and Bonferroni correction, was attained at *p* < 0.05 (*) (**: *p* < 0.005, ***: *p* < 0.0005, ****: *p* < 0.00005, ns: no significant difference).

**Figure 5 biomedicines-08-00251-f005:**
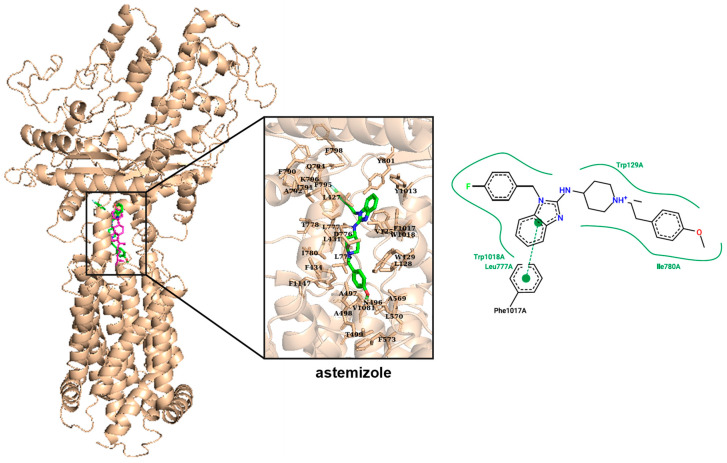
Docking of astemizole in the human Ptch1 protein structure. Chain A from pdb 6n7h is represented on the left. Cholesterol (magenta) and astemizole (green) bind in the same central binding cavity when docked with Vina. Astemizole interacts with similar amino acids as cholesterol (see Table 2). On the right, 2D representation of interactions between astemizole and Ptch1: curved green line represents hydrophobic interactions, and dashed green line represents pi-stacking interactions.

**Figure 6 biomedicines-08-00251-f006:**
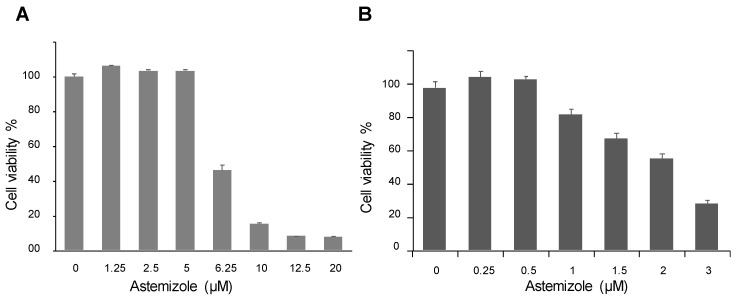
Astemizole has a cytotoxic and anti-proliferative effect on ACC cells. Cell viability was measured after treatment with a serial dilution of astemizole added on H295R cells at 80% confluence for 48 h (**A**), or added for 7 days when H295R cells were plated (**B**). The graphs reported are representative of more than 3 repeated experiments.

**Figure 7 biomedicines-08-00251-f007:**
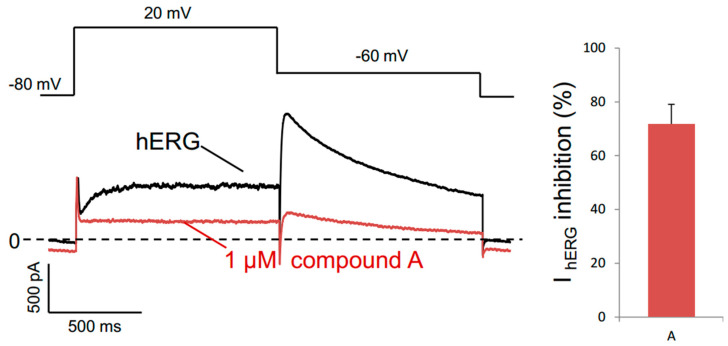
Effect of astemizole on hERG. I_hERG_ “pulse” amplitude was monitored on HEK293 cells overexpressing hERG during repetitive application of the protocol shown in the upper panel. Currents in control (black) and the presence of 1 μM astemizole (red) have been recorded. The percentage of reduction in I_hERG_ pulse amplitude induced by 1 μM astemizole is reported.

**Table 1 biomedicines-08-00251-t001:** Astemizole increases the cytotoxicity of chemotherapeutic treatments on adrenocortical carcinoma (ACC) cells. Cell viability was measured after 48 h of treatment of H295R cells with serial dilutions of doxorubicin (dxr) alone or in combination with 10 μM mitotane and 10 μM etoposide (EDM) in the presence or the absence of 1 μM astemizole. IC_50_ values of dxr reported are the mean ± SEM of 3 independent experiments.

	Dxr-IC_50_ (µM)	
Treatment	Without Astemizole	With Astemizole
Dxr	17.3 ± 3.6	1.65 ± 0.4
EDM	15.93 ± 4	1.84 ± 0.7

**Table 2 biomedicines-08-00251-t002:** List of amino acids within a radius of 6 Å from cholesterol, doxorubicin or astemizole.

Ligand	Amino Acid Involved
Cholesterol	V125, E126, **L128**, W129, *L427*, *L431*, F434, N496, *A497*, A498, **T499**, V502, *I567*, *A569*, L570, *F573*, L775, **D776**, L777, I780, Q794, Y801, F987, **Y1013**, F1017, **W1018**, **Q1020**, S1079, **V1081**, F1147, I1148, Y1151, **F1152**
Astemizole	*V125*, ***L128***, *W129*, **L427**, **L431**, *F434*, *N496*, **A497**, *A498*, ***T499***, **A569**, *L570*, *L775*, ***D776***, *L777*, *T778*, *D779*, *I780*, F790, I791, A792, *Q794*, F795, K796, F798, *Y801*, ***Y1013***, *F1017*, ***W1018***, *V1081*, *F1147*
Doxorubicin	N124*, V125, L128, W129,* F422*,* T424*, **L427**, **L431**,* G774*,* L775*,* **D776***,* L777*,* F1017*,* **W1018***,* Y1021, S1079

Underlined have side chains toward the cholesterol, in **bold** when mutation results in damaging phenotype, *x* when conserved in the Patched family, in *italic* if common with the ones listed for the cholesterol, in red if common with the ones listed for the doxorubicin.

**Table 3 biomedicines-08-00251-t003:** Profiling data for astemizole.

LogD7.4	Aqueous Solubility (µM)	RLM T1/2 (min)	RLM (µL/min/mg)	HLM T1/2 (min)	HLM (µL/min/mg)	Ppb (%)
3.5	22.1	31	< 115.5	> 60	< 115.5	99

The distribution coefficient between octanol and phosphate buffer saline (PBS) measured at pH 7.4 (Log D 7.4), the solubility in aqueous buffer (PBS, pH 7.4) and propylene glycol (PG), the chemical stability in aqueous solution (PBS, pH 7.4) and PG over 7 days, the metabolic stability in human and rat liver microsomes (HLM and RLM, respectively) and the human plasma protein binding (ppb %) are reported.

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
