# Peer review of "Astemizole Sensitizes Adrenocortical Carcinoma Cells to Doxorubicin by Inhibiting Patched Drug Efflux Activity"

_biomedicines, 2020, doi:10.3390/biomedicines8080251_

Round 1

Reviewer 1 Report

The paper by  Anida Hasanovic et al., "Astemizole sensitizes adrenocortical carcinoma cells to doxorubicin by inhibiting Patched drug efflux activity"  seeks to increase Adrenocortical carcinoma treatment efficacy by screening for a molecule to add to the ACC  standard of care.

The authors claim that their data provide evidence that the use of astemizole or a derivative in combinations with EDPM could be a promising therapeutic option for ACC by increasing the treatment effectiveness at lower doses of EDPM which would reduce the severe side effects of this regimen.

Indeed this is an important question,  however,  they present vast in-vitro data, the most important part (missing) is in-vivo work... to validate the data, not just Pharmacokinetic study (where is that data)?.

Language can be improved(Grammar)

Author Response

Reviewer 1 comments:

The paper by Anida Hasanovic et al., "Astemizole sensitizes adrenocortical carcinoma cells to doxorubicin by inhibiting Patched drug efflux activity" seeks to increase Adrenocortical carcinoma treatment efficacy by screening for a molecule to add to the ACC  standard of care.

The authors claim that their data provide evidence that the use of astemizole or a derivative in combinations with EDPM could be a promising therapeutic option for ACC by increasing the treatment effectiveness at lower doses of EDPM which would reduce the severe side effects of this regimen.

Indeed this is an important question however, they present vast in-vitro data, the most important part (missing) is in-vivo work... to validate the data, not just Pharmacokinetic study (where is that data)?

Our response:

Our study shows that astemizole increases doxorubicin efficacy against adrenocortical carcinoma cells and melanoma cells in vitro by inhibiting the doxorubicin efflux activity of Patched. We previously showed that these cancer cell lines endogenously express Patched, and that Patched participates to the efflux of doxorubicin in these two cell lines (Hasanovic et al, 2018; Signetti et al. 2020). This study allowed us to identify astemizole as a new inhibitor of Patched drug efflux, and Patched as a new target of astemizole. However, astemizole was withdrawn from the market in 1999 because of its high affinity for the potassium cardiac channel hERG (as reported in figure 7) and potential adverse cardiac reactions. Therefore, there is no interest to test astemizole on ACC or melanoma xenografts in mice since this compound will never be considered as an adjuvant to chemotherapy treatment in a clinical assay. We are currently working on the chemical modification of astemizole to make it lose its affinity for hERG. The optimized lead obtained will be tested on mice grafted with ACC cells or melanoma cells to evaluate in vivo activity of this lead.

Language can be improved(Grammar)

Our response:

A professional English editor reviewed the revised manuscript’s text.

Reviewer 2 Report

Dear Sirs

The paper by Hasanovic et al., identified a new Patch inhibitor and demonstrated an increased cytotoxicity using human adrenocortical carcinoma in vitro. I think this story could be correct if authors follow some standards to describes new phenomenon. At this point there are 2 major flaws:

  1. The founded concentration of astemizole was detected in 1 cancer cell lines and NO negative control ( such as fibroblasts have been used). Additionally, using one cell lines made it impossible to extrapolate the results for other cell including primary
  2.  No Chou-Talalay method was applied to obtain the optimal concentration for astemizole+ dox or astemizole+EDM. As such the detected synergy can be suboptimal
  3.  Figure 3D. Increasing caspase activation from 2% to 12% can statistical significant but can it translated to biological significance in vivo?

Author Response

Reviewer 2 comments:

The paper by Hasanovic et al., identified a new Patch inhibitor and demonstrated an increased cytotoxicity using human adrenocortical carcinoma in vitro. I think this story could be correct if authors follow some standards to describes new phenomenon. At this point there are 2 major flaws:

1. The founded concentration of astemizole was detected in 1 cancer cell lines and NO negative control (such as fibroblasts have been used). Additionally, using one cell lines made it impossible to extrapolate the results for other cell including primary

Our response:

We have performed our experiments on the adrenocortical carcinoma cell line NCI-H295R which is the human differentiated ACC cell line of reference to study ACC, where we demonstrated that Patched strongly participates to the doxorubicin efflux and doxorubicin resistance in this cell line (Hasanovich et al., 2018). We could also perform experiments on the very recently established MUC-1 cell line from Hantel at al. Oncotarget 2016 but we did not demonstrate the role of Patched in the efflux of doxorubicin of this ACC cell line yet. We don't have a normal adrenocortical cell line available. However, we tested the effect of astemizole on the human melanoma cell line MeWo for which we showed an endogenous expression of Patched and demonstrated the role of Patched in dxr efflux (Signetti et al. 2020), and on the non-tumorigenic human keratinocyte cell line HaCaT. We showed that astemizole increases the cytotoxic effect of doxorubicin against MeWo cells, and has only slightly cytotoxic effect on non-tumorigenic keratinocyte cells. Those results have been added in the revised version of the manuscript and in Supplementary Fig. 2.

2. No Chou-Talalay method was applied to obtain the optimal concentration for astemizole+ dox or astemizole+EDM. As such the detected synergy can be suboptimal

Our response:

The Fa-CI plot and normalized isobologram analysis were performed as described in (Kroiss et al. 2016) using the Chou-Talalay’s method for drug combination and Compusyn Software (Version 1.0 downloaded from www.combosyn.com, Chou et al. 2006), and reported in the supplementary figure 1. This analysis clearly shows that dxr and astemizole exhibit significant synergism as demonstrated by cooperativity indices far below 1. The synergy quantification using the Chou-Talalay method was described in the Materials and Methods section and reported in the results section of the revised manuscript version.

3. Figure 3D. Increasing caspase activation from 2% to 12% can statistical significant but can it translated to biological significance in vivo?

Our response:

Caspase activation of cells was reported as a factor of the caspase activity of the control cells treated with DMSO in Fig. 3B. The results show that the use of 0.5 µM astemizole does not increase caspase 3/7 activation by itself, but multiplies by 5 caspase activation relative to cells treated with 2 µM dxr without astemizole. A factor of 5 in the number of apoptotic cells should have a biological significance in vivo. This was not clearly explained on the figure and results. We thus modified the title of the y-axis of Fig. 3B, the legend of this figure, and the text in the results section to make these apoptosis results clearer.

Round 2

Reviewer 1 Report

Please discuss the in-vivo issue.

Author Response

We’ve added the following sentences in the discussion (line 7 page 12) regarding in vivo study as requested by Reviewer 1: “We are currently working on the chemical modification of astemizole to make it lose its affinity for hERG. The optimized compound obtained will be tested on mice grafted with ACC cells to evaluate in vivo activity of this lead.”

Reviewer 2 Report

All comments are being addressed

Author Response

Thank you very much for reviewing our manuscript.